# Limb Length Discrepancy and Corticospinal Tract Disruption in Hemiplegic Cerebral Palsy

**DOI:** 10.3390/children9081198

**Published:** 2022-08-10

**Authors:** Hyo Sung Kim, Su Min Son

**Affiliations:** Department of Physical Medicine and Rehabilitation, College of Medicine, Yeungnam University, Daegu 42415, Korea

**Keywords:** hemiplegia, cerebral palsy, corticospinal tract, diffusion tensor, limb length, motor, growth

## Abstract

This study aimed to investigate the relationship between the corticospinal tract (CST) and limb length discrepancy (LLD) in patients with hemiplegic cerebral palsy (CP). Using diffusion tensor tractography, a retrospective study on 92 pediatric patients with hemiplegic CP who visited our hospital from May 2017 to the end of 2020 was conducted. Limb length was measured by anthropometry to calculate LLD. The functional level of hemiplegia scale (FxL), modified Ashworth scale, and manual muscle test (MMT) were evaluated for clinical function. Patients were classified into two groups according to the presence or absence of disruption of the affected CST: disruption (A) and preservation (B) groups. Fractional anisotropy (FA) and mean diffusivity (MD) of the affected CSTs were measured and correlated with LLD. The results of the independent *t*-test and chi-square test did not show significant differences between the two groups, except in the FxL and finger extensor of MMT (*p* < 0.05). For the LLD, there were no significant differences in total upper, total lower, and foot limb lengths. A significant difference was observed only in hand LLD (*p* < 0.05) from ANCOVA. Hand LLD was significantly correlated with FA (r = −0.578), MD (r = 0.512), and degree of CST disruption (r = −0.946) from the Pearson correlation test. The results of this study suggested that patients with hemiplegic CP would likely have LLD especially in the hand, and that CST evaluation using diffusion tensor tractography might be helpful in assessing and predicting LLD in hemiplegic CP.

## 1. Introduction

Cerebral palsy (CP) is a syndrome characterized by abnormal movement and posture in patients with non-progressive immature brain lesions. Among patients with several types of CP, patients with hemiplegia show asymmetry in motor performance and right and left limb lengths. Limb length discrepancy (LLD) can be observed in pediatric patients with hemiplegic CP, even in toddlers, and is known to increase with age [1]. LLD can cause various symptoms in pediatric patients under growth. At a young age, there exists only a difference between right and left limb lengths. However, as the age increases, LLD can lead to other problems, such as scoliosis, abnormal limping gait, musculoskeletal problem, and pain. Therefore, early diagnosis is necessary, and it is very important to evaluate the factors that are related to LLD. There have been a few studies on LLD in patients with hemiplegic CP [2,3,4]. Uvebrant et al. reported that 96% of patients with hemiplegic CP showed a mean discrepancy and that LLD was more prominent in the upper limb (15 mm) than in the lower limb (6 mm) [2]. Zonta also reported growth impairment in the affected limbs in patients with hemiplegic CP and that LLD was more prominent in the distal parts, such as the hand and foot, than in the total upper and lower limbs [4]. The results of these studies implied that LLD is more prominent in the upper limb than in the lower limb, and distal limb rather than the proximal limb. Consequently, there is a high possibility that LLD is prominent in the hand rather than in the foot or total upper or total lower limb. Previous studies on LLD used anthropometry for measurement of the length of the hand, foot, arm, or leg. Recent studies have attempted to confirm the relationship between bone length, bone diameter, or bone age and LLD using radiography or magnetic resonance imaging (MRI) [5,6,7,8]. However, the cause of CP is brain pathology, so it is assumed that an approach to the causal relationship between brain pathology and LLD is necessary.

It is well known that the corticospinal tract (CST) is mainly related to motor function, especially hand function [9,10,11]. The CST is composed of the lateral CST, which is formed by approximately 90% of CST fibers, and the anterior CST, which is formed by approximately 10% of CST fibers [12]. The lateral CST is related to the distal musculature for fine motor function, particularly that of the hands, whereas the anterior CST is related to the axial and proximal motor function [11]. Therefore, lateral CST can affect limb growth, particularly in growing pediatric patients.

Diffusion tensor tractography (DTT), derived from diffusion tensor imaging (DTI), has poor temporal resolution but excellent spatial resolution. Among the techniques currently used for brain evaluation, DTT is the only modality to enable three-dimensional visualization of subcortical neural tracts in vivo. Moreover, quantitative information on the white matter status can be provided by diffusion parameters, such as fractional anisotropy (FA) and mean diffusivity (MD). It is possible to evaluate white matter tracts even in patients whose brain myelination has not been completed, such as infants, and various neural tracts with a single conduct of DTI. Several previous studies have demonstrated that DTT is very helpful for CST assessment [9,10,11,12,13]. However, the causal relationship between LLD and CST has not yet been studied. Therefore, we attempted to confirm the association between LLD and CST using DTT and to confirm the usefulness of DTT in the assessment of LLD in patients with hemiplegic CP.

## 2. Materials and Methods

### 2.1. Patients

Ninety-two pediatric patients (mean age, 7.03 ± 2.56 years; range, 2–12 years) were retrospectively recruited according to the following criteria from May 2017 to the end of 2020: (1) diagnosis of hemiplegic CP by two pediatric neurologists; (2) diagnosis of hemiplegic CP for at least 1 year; (3) Limb length measurement performed within 1 month of DTI scanning; (4) absence of congenital anomaly, fracture, and operation history on extremities; and (5) no history of botulinum toxin injection on extremities within the previous 6 months. Patients diagnosed with genetic syndromes, malnutrition, peripheral neuropathy, or endocrinopathies were excluded. The study population was originally selected from 196 pediatric patients with hemiplegic CP aged 2–12 years. Of these 196 patients, 113 who underwent limb length measurement within 1 month of DTT scanning were selected. Of these 113 patients, seven were excluded due to a history of botulinum toxin injection within the previous 6 months. Twelve patients with a history of orthopedic surgery of the extremities were excluded. Two patients with genetic syndromes were excluded. The remaining 92 patients were enrolled in the study (Figure 1). All participants were diagnosed with spastic hemiplegic CP. To reflect the difference in growth-related LLD, the patients who could walk, even with the use of orthoses or walker, were enrolled in the study. Patients with preterm birth < 37 weeks, extremely low birth weight < 1000 g, or brain lesion on conventional MRI were also included. An explanation sheet was provided in consideration of the characteristics of the pediatric patient, and if the patient did not understand the explanation, explanation and informed consent statements were provided to the guardians. All examination procedures and possible side effects and safety issues were explained before the examination. Pediatric neurologists and guardians accompanied the patient throughout the entire process of the study, observing and monitoring the patient’s condition. When the patient’s condition was unfavorable or safety-related issues were suspected, the study was stopped or delayed for the patient. However, none of the 92 patients stopped the test or showed any serious safety-related issues. The pediatric neurologists were unware of the DTT results. Two analysts of DTT (Son SM and Kim HS) were also unaware of the clinical information. They analyzed the DTT in a blind state to the clinical information before the statistical analysis was completed. Informed consent was obtained from the parents of all participants, and the institutional review board of our hospital approved the study protocol.

### 2.2. Clinical Evaluation

Limb length was measured using anthropometry. This method is widely used in clinical settings because it is easily applicable, noninvasive, and inexpensive and does not require irradiation [14,15]. The total upper limb, hand, total lower limb, and foot lengths were measured according to the standardized measurement rules as follows: total upper limb length was measured from the most superior lateral point of the acromion process (acromial landmark) to the lower and lateral border of the styloid process of the radius (radial landmark). Hand length was measured from the styloid process at the base of the thumb to the tip of the middle finger, with the hand extended and palm rested in the direction of the longitudinal axis of the forearm. The total lower limb length was measured from the anterior superior iliac spine to the medial malleoli of the ankle (from one fixed bony point to another). Foot length was measured from the most posterior part (center) of the heel to the most anterior part of the longest toe (second toe) [15]. The discrepancy between both sides was calculated as a percentage by comparing the affected and unaffected sides using the following equation: (100 × (value of the unaffected side − value of the affected side)/value of the unaffected side).

Functional level of hemiplegia scale (FxL) was used to evaluate the functional level of the hemiplegic extremities. The FxL is a widely used measurement tool in pediatric patients with hemiplegic CP [16,17]. FxL classifies hemiplegic extremities as follows: 0, no use; 1, use as a stabilizing weight only; 2, can hold objects placed in hand; 3, can hold objects and stabilize for use in the other hand; 4, can actively grasp object and hold it weakly; 5, can actively grasp object and stabilize well; 6, can actively grasp object and manipulate it against other hand; 7, can easily perform bimanual activities and occasionally use the hand spontaneously; and 8, can use the hand with complete independence. We also used the modified Ashworth scale (MAS) and manual muscle test (MMT) to assess spasticity and muscle strength of the upper and lower extremities on the affected side [18,19]. The unaffected side was the dominant side in all patients.

Two pediatric neurologists examined the patients independently twice with 1-week interval between examinations, and all patients were examined twice by two neurologists. The results of FxL, MAS, and MMT by two neurologists were the same in all patients. The consistent rate of LLD was 96% for the inter-observer’s analysis and 97% for the intra-observer’s analysis when lengths < 0.5 cm are truncated and limb length was measured in centimeters. These results showed a high reliability and validity rate.

### 2.3. Diffusion Tensor Tractography

DTI data were acquired using a six-channel head coil on a 1.5-T Philips Gyroscan Intera (Philips, Best, the Netherlands) with single-post echo-planar imaging. We acquired 60 contiguous slices parallel to the anterior—posterior commissure line for each of the 32 non-collinear diffusion sensitizing gradients. Imaging parameters were as follows: acquisition matrix, 96 × 96; reconstructed matrix, 128 × 128; field of view, 221 mm × 221 mm; repetition time, 10,726 ms; time to echo, 76 ms; parallel imaging reduction factor (SENSE factor), 2; echo-planar imaging factor, 67; b, 1000 s/mm^2^; number of excitations, 1; slice thickness, 2.3 mm (acquired isotropic voxel size = 2.3 mm × 2.3 mm× 2.3 mm). DTI data were analyzed using the Oxford Centre for Functional Magnetic Resonance Imaging of the Brain Software Library (FSL; www.fmrib.ox.ac.uk/fsl, 8 March 2021). Affine multi-scale two-dimensional (2D) registration was used to correct the head motion effect and image distortion due to eddy currents. Fiber Assignment by Continuous Tracking, a three-dimensional fiber reconstruction algorithm in Philips PRIDE software (Philips Medical Systems, Best, the Netherlands), was used to evaluate fiber connectivity.

CST fiber tracking was performed using a fractional anisotropy (FA) threshold of >0.2 and direction threshold of 60°. In each case, on a 2D FA color map, a seed region of interest (ROI) was drawn in the CST portion of the anterior mid-pons, and another ROI was drawn in the CST portion of the anterior low-pons. Fiber tracts passing through both ROIs were designated as final tracts of interest. Patients were classified into two groups according to the presence or absence of affected CST disruption: CST disruption group (group A) and CST preservation group (group B) (Figure 2). The degree of CST disruption was defined as follows: 1, CST disruption at the pons level; 2, CST disruption at the subcortex level; and 3, CST preservation at the cerebral cortex level. FA and MD of the depicted CSTs were measured.

To measure the inter- and intra-observer’s variation, the DTT analysis was performed blindly and independently by two authors (Son SM and Kim HS). Each author analyzed the data of each patient twice, randomly, with 1-week interval between examinations. The DTI data of each subject were included as DTT data when the degree of CST disruption result of each analyzer was the same. All results were the same, and all data from the 92 subjects were included as DTT data. The consistent rate of the inter-observer’s analysis was 97%. The consistent rate of the intra-observer’s analysis was 98%, demonstrating high reproducibility rate.

### 2.4. Statistical Analyses

SPSS software (version 18.0; SPSS, Chicago, IL, USA) was used for data analysis. Independent *t*-test and chi-square test were used to determine statistical differences between groups A and B in demographic data. Independent *t*-test and ANCOVA were used to determine the difference in LLD. The Pearson correlation test was used to determine the statistical significance of the correlation between the LLD and DTI parameters. Statistical significance was set at a *p*-value < 0.05.

## 3. Results

Table 1 summarizes patients’ demographic and clinical data. Of the 92 patients, 47 were assigned to group A and 45 to group B. No significant differences in demographic or clinical data were observed between the two groups, except for the FxL and affected finger extensor of the MMT and degree of CST disruption (*p* < 0.05).

The degree of LLD between the affected and unaffected sides is shown in Table 2. A significant difference was observed only in hand length (*p* < 0.05), not in total upper limb length, total lower limb length, and foot length (*p* > 0.05) from the ANCOVA. When CST disruption was severe, decreased FA and increased MD were observed, the correlation between these DTI parameters and the degree of LLD showed significant results with FA (*r*= −0.578) and MD values (*r*= 0.512) only in hand length discrepancy. Total upper, total lower, and foot length discrepancy showed no significant results. The correlation between the degrees of LLD and the degree of CST disruption also revealed no significant results in upper total limb length, lower total limb length, and foot length; however, correlations in the hand length discrepancy (*r* = −0.946) were statistically significant (Table 3).

## 4. Discussion

This study demonstrated a significant difference in hand length but not in the total upper limb length, total lower limb length, or fool length in hemiplegic CP. This hand length discrepancy showed a significant correlation with the DTI parameters of the CST. Our results are in agreement with the findings of previous studies, in which LLD was more prominent in the distal upper limb than proximal or lower limb.

Although there are several hypotheses regarding the cause of LLD in hemiplegic CP, it has not yet been clearly elucidated [1,3,4,20]. Demir et al. reported that brain damage before early childhood can influence the growth of the affected long bones because of decreased vascularity, hormonal changes, lack of mechanical forces, and nutrition on the affected side [1]. Another study by Stevenson et al. reported that non-nutritional factors related to disease severity have a significant influence on growth [3]. Different results have been observed for spasticity, which is one of the causes of LLD. Lin et al. reported that spasticity causes mechanical stress on the bone and is beneficial to bone growth in hemiplegic CP [21]. However, Zonta et al. reported that increased spasticity was associated with increased LLD [20]. Another study reported that the level of spasticity was not an important factor in LLD [1]. As described above, there is no established cause of LLD in children with hemiplegic CP. However, previous studies have reported that central nervous system injury can affect musculoskeletal maturation either directly or indirectly [1,4,22].

Our results showed a significant correlation between hand length discrepancy and affected CST status. The FA value indicates the degree of directionality and integrity of white matter microstructures, such as axons, myelin, and microtubules. The MD value indicates the magnitude of water diffusion, which can increase under conditions of vasogenic or cytotoxic edema or accumulation of cellular debris from axonal injury [23]. Therefore, the decreased FA and increased MD values indicated disintegration of the CST fibers. Moreover, the observed correlation between the degrees of hand length discrepancy and CST disruption suggests that more severe disintegration of the CST is related to a greater discrepancy in hand length.

In conclusion, the corticospinal tract is thought to have meaningful relevance with distal limb length discrepancy, especially with hand length in patients with hemiplegic cerebral palsy. Therefore, if patients with hemiplegia had severe disruption of CST, close monitoring for the possibility of limb length discrepancy is recommended. To the best of our knowledge, this study is the first to demonstrate the relationship between LLD and CST in hemiplegic CP using DTT. However, this study has several limitations. First, additional imaging evaluations, such as radiography, were not performed. Second, additional electrophysiological evaluations, including electromyography/nerve conduction velocity and motor-evoked potential tests, were not performed. Third, more detailed identical functional evaluations were not performed due to the wide age range of the patients. Fourth, evaluations on the various possibilities of LLD by age and different causes of hemiplegia have not been conducted. Another limitation was lack of healthy control group, so the comparison of LLD between the healthy controls and CP was not evaluated. Finally, the limitations of DTI should be considered. DTI may underestimate or overestimate the neural fiber tract because of the region of fiber complexity. Crossing can prevent the full reflection of the underlying fiber architecture using DTI. Besides, the probabilistic technique using FSL can provide more accurate information about the presence of branching fibers or preservation of weak white matter integrity than the deterministic technique using PRIDE or DTI studio, which was used in this study [24]. Therefore, complementary probabilistic DTI studies with larger case numbers that include participants with different causes of hemiplegia, various ages, and normal healthy subjects and detailed and identical clinical, radiological, and electrophysiological evaluations are warranted.

## Figures and Tables

**Figure 1 children-09-01198-f001:**
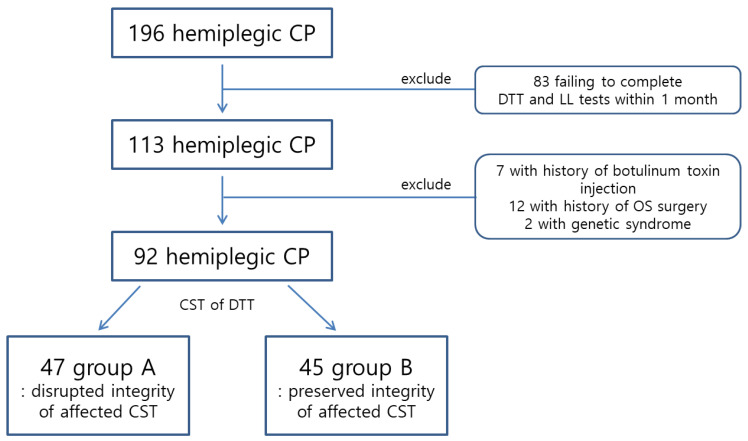
Flow chart of study design. (CP, cerebral palsy; CST, corticospinal tract; DTT, diffusion tensor tractography; LL, limb length evaluation; OS, orthopedic surgery).

**Figure 2 children-09-01198-f002:**
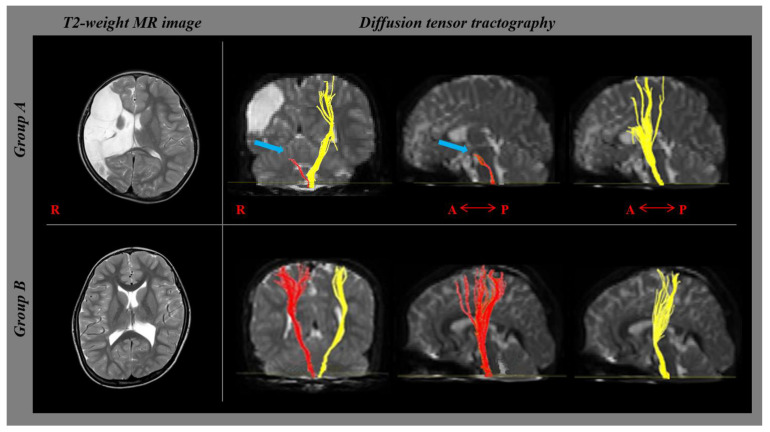
Diffusion tensor tractography of the corticospinal tract in patients. (**A**) shows the disruption of the right corticospinal tract (blue arrow) in patient who were diagnosed as left hemiplegic cerebral palsy. (**B**) shows preservation of both corticospinal tracts to the cortex level, although the left one has mildy decreased fiber volume compared to right one. Patient B was diagnosed with right hemiplegic cerebral palsy.

**Table 1 children-09-01198-t001:** Demographic and clinical data of the patients.

		Group A (n = 47)	Group B (n = 45)	*p*-Value	Difference of 95% CI ^#^
Male		26	25	0.982 *	
Mean age, yr (SD)		7.12 ± 3.12	6.97 ±3.84	0.780	1.211/−0.911
Affected side, Right		25	20	0.401 *	
Degree of CST disruption (1/2/3)		26/21/- *	-/-/45 *	<0.001 *	
FxL		4.91 ± 0.69	5.8 ± 0.58	<0.001	−0.477/−1.293
MMT	Shoulder abductor	3.62 ± 0.39	3.78 ± 0.40	0.053	0.002/−0.325
	Finger extensor	3.08 ± 0.33 *	4.31 ± 0.45 *	<0.001	−1.100/−1.352
	Hip abductor	4.05 ± 0.42	4.58 ± 0.39	0.050	0.687/−0.383
	Ankle dorsiflexor	3.96 ± 0.58	4.15 ± 0.78	0.062	0.063/−0.312
MAS	Shoulder adductor	1.07 ± 0.66	0.98 ± 0.52	0.491	0.332/−0.160
	Finger flexor	1.67 ± 0.55	1.45 ± 0.43	0.052	0.349/−0.080
	Hip adductor	0.94 ± 0.42	0.95± 0.54	0.94	0.220/−0.238
	Ankle plantar flexor	1.18 ± 0.71	1.10± 0.53	0.425	0.281/−0.119

Values represent the mean ± SD. * indicates the results of chi-square test. ^#^ marked with upper limit/lower limit.

**Table 2 children-09-01198-t002:** Limb length discrepancy (%) between affected side unaffected side in two groups.

	Group A	Group B			
	Discrepancy (%)		P1	Difference of 95% CI	P2
Total upper limb	1.72 ± 1.94	1.51 ± 1.86	0.013 *	0.375/0.045	0.099
Hand	7.27 ± 3.04	2.65 ± 2.86	<0.001 *	5.052/4.195	0.030 *
Total lower limb	1.35 ± 2.29	1.26 ± 2.60	0.344	0.279/−0.098	0.752
Foot	3.51 ± 4.58	2.86 ± 3.46	<0.001 *	1.023/0.278	0.208

Values represent mean ± SD. * *p* < 0.05. P1 denotes the results of independent *t*-test. P2 denotes the results of one way ANCOVA.

**Table 3 children-09-01198-t003:** Correlation of limb length discrepancy (%) with DTI parameters and with the degree of CST disruption.

	Correlation Coefficient (r)		
	FA	MD	Degree of CST Disruption
Total upper limb	−0.065	0.096	−0.146
Hand	−0.578 *	0.512 *	−0.946 *
Total lower limb	−0.092	0.065	−0.132
Foot	−0.132	0.194	−0.186

Values represent mean ± SD. * *p* < 0.05. FA: fractional anisotropy, MD: mean diffusivity, CST: corticospinal tract.

## Data Availability

All data are provided in the paper.

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
