# Peer review of "Limb Length Discrepancy and Corticospinal Tract Disruption in Hemiplegic Cerebral Palsy"

_children, 2022, doi:10.3390/children9081198_

Round 1

Reviewer 1 Report

Thank you for giving me this opportunity to review this article. The article is well written, though I have some serious concerns regarding the article.

Abstract:

  1. Include the objective of the study.
  2. Mention the type of study design.
  3. Mention the study duration and study setting.
  4. Include the study outcome measures.
  5. Mention the statistical tests used for the study.
  6. The results should be presented with 95%CI (upper limit – lower limit) for all the variables.
  7. Avoid abbreviations in the conclusion.
  8. The conclusion should be concise and self-explanatory and drawn on the basis of study reports.

Manuscript

  1. Include more information about DTT, It’s application procedures, merits and demerits?
  2. The novelty of the study is missing, including more recent references emphasizing the need for this study.
  3. The study objective is missing.
  4. Include the clinical significance of this study over clinicians, patients, and researchers after the study hypothesis.
  5. Follow the strict author guidelines to present the paper. (CONSORT).
  6. Mention the type of study design.
  7. Mention the study duration and study setting.
  8. Include the character of the study participants in detail.
  9. Include the ethical and clinical trial registration number.
  10. Include the randomization and allocation of subjects in detail.
  11. How the safety concerns are measured?
  12. Include the study outcome measures and its reliability and validity.
  13. Include the information about the blinding procedures. 
  14. Include the reference study for sample size calculation.
  15. The statistical tests used for the study was not apt to this study.
  16. The results should be presented with 95%CI (upper limit – lower limit) for all the variables.
  17. Avoid abbreviations in the conclusion.
  18. The conclusion should be more concise and self-explanatory and drawn on the basis of study reports. 
  19. Add more real-time limitations faced by the researcher and the study. 
  20. Include future recommendations of the study.

Author Response

Thank you for giving me this opportunity to review this article. The article is well written, though I have some serious concerns regarding the article.

Abstract:

  1. Include the objective of the study.
  2. Mention the type of study design.
  3. Mention the study duration and study setting.
  4. Include the study outcome measures.
  5. Mention the statistical tests used for the study.
  6. The results should be presented with 95%CI (upper limit – lower limit) for all the variables.
  7. Avoid abbreviations in the conclusion.
  8. The conclusion should be concise and self-explanatory and drawn on the basis of study reports.

Answer: I appreciate your comment. I revised abstract according to your comment.

Abstract:

This study aimed to investigate the relationship between the corticospinal tract (CST) and limb length discrepancy (LLD) in patients with hemiplegic cerebral palsy (CP). Using diffusion tensor tractography, a retrospective study on 92 pediatric patients with hemiplegic CP who visited our hospital from May 2017 to the end of 2020 was conducted. Limb length was measured by anthropometry to calculate LLD. The functional level of hemiplegia scale (FxL), modified Ashworth scale, and manual muscle test (MMT) were evaluated for clinical function. Patients were classified into two groups according to the presence or absence of disruption of the affected CST: disruption (A) and preservation (B) groups. Fractional anisotropy (FA) and mean diffusivity (MD) of the affected CSTs were measured and correlated with LLD. The results of the independent t-test and chi-square test did not show significant differences between the two groups, except in the FxL and finger extensor of MMT (p<0.05). For the LLD, there were no significant differences in total upper, total lower, and foot limb lengths. A significant difference was observed only in hand LLD (p<0.05) from one way ANCOVA. Hand LLD was significantly correlated with FA (r=-0.578), MD (r=0.512), and degree of CST disruption (r=-0.946) from the Pearson correlation test. The results of this study suggested that patients with hemiplegic CP would likely have LLD especially in the hand and that CST evaluation using diffusion tensor tractography might be helpful in assessing and predicting LLD in hemiplegic CP.

Manuscript

  1. Include more information about DTT, It’s application procedures, merits and demerits?

Answer: Thank you for your valuable comments. I added information about DTT as follows.

Diffusion tensor tractography (DTT), derived from diffusion tensor imaging (DTI), has poor temporal resolution but excellent spatial resolution. Among the techniques currently used for brain evaluation, DTT is the only modality to enable three-dimensional visualization of subcortical neural tracts in vivo. Moreover, quantitative information on the white matter status can be provided by diffusion parameters, such as fractional anisotropy and mean diffusivity. It is possible to evaluate white matter tracts even in patients whose brain myelination has not been completed, such as infants, and various neural tracts with a single conduct of DTI. Several previous studies have demonstrated that DTT is very helpful for CST assessment.

  1. The novelty of the study is missing, including more recent references emphasizing the need for this study.

Answer: Thank you for the comment. I revised the manuscript and added the recent references as follows.

There have been a few studies on LLD in patients with hemiplegic CP [2-4]. Uvebrant et al. reported that 96% of patients with hemiplegic CP showed a mean discrepancy, and that LLD was more prominent in the upper limb (15 mm) than in the lower limb (6 mm) [2]. Zonta also reported growth impairment in the affected limbs in patients with hemiplegic CP, and that limb length discrepancy was more prominent in distal parts, such as the hand and foot, than in the total upper and lower limbs [4]. The results of these studies implied that LLD is more prominent in the upper limb than in the lower limb, and distal limb rather than the proximal limb. Consequently, there is a high possibility that LLD is prominent in the hand rather than in the foot or total upper or lower limb. Previous studies on LLD using anthropometry for measurement of the length of the hand, foot, arm, or leg. Recent studies have attempted to confirm the relationship between bone length, bone diameter, or bone ageand LLD using radiograph or magnetic resonance imaging (MRI) [5-8]. However, the cause of CP is brain pathology, so it is assumed that an approach to the causal relationship between brain pathology and LLD is necessary.

[7]   Schroeder KM, Heydemann JA, Beauvais DH. Musculoskeletal Imaging in Cerebral Palsy. Phys Med Rehabil Clin N Am. 2020 Feb;31(1):39-56.

[8]  Lee JS, Choi IJ, Shin MJ, Yoon JA, Ko SH, Shin YB. Bone age in unilateral spastic cerebral palsy: is there a correlation with hand function and limb length? J Pediatr Endocrinol Metab. 2017 Mar 1;30(3):337-341.

  1. The study objective is missing.

Answer: Thank you for your comment, I revised the introduction section following your comment.

Several previous studies have demonstrated that DTT is very helpful for CST assessment [9-13]. However, the causal relation between LLD and CST has not yet been studied. Therefore, we attempted to confirm the association between LLD and CST using DTT and to confirm the usefulness of DTT in the assessment of LLD in patients with hemiplegic CP.

  1. Include the clinical significance of this study over clinicians, patients, and researchers after the study hypothesis.

Answer: I appreciate your valuable comments.

Cerebral palsy (CP) is a syndrome characterized by abnormal movement and posture in patients with non-progressive immature brain lesions. Among patients with several types of CP, patients with hemiplegia show asymmetry in motor performance and right and left limb lengths. Limb length discrepancy (LLD) can be observed in pediatric hemiplegic CP, even in toddlers, and is known to increase with age [1]. LLD can cause various symptoms in pediatric patients under growth. At a young age, there exists only a difference between right and left limb lengths. However, as the age increases, LLD can lead to other problems, such as scoliosis, abnormal limping gait, musculoskeletal problem, and pain. Therefore, early diagnosis is necessary, and it is very important to evaluate the factors that are related to LLD. There have been a few studies on LLD in patients with hemiplegic CP [2-4].

  1. Follow the strict author guidelines to present the paper. (CONSORT).

Answer: Sorry for this. I added the information in the text.

Author Contributions: Conceptualization, H.-S.K. and S.-M.S. ; methodology, H.-S.K. and S.-M.S. ; software, H.-S.K. ; validation, H.-S.K. and S.-M.S.; formal analysis, H.-S.K. ; investigation, H.-S.K. and S.-M.S. ; resources, H.-S.K. and S.-M.S.; data curation, H.-S.K. and S.-M.S. ; writing—original draft preparation, H.-S.K. and S.-M.S. ; writing—review and editing, H.-S.K. and S.-M.S. ; visualization, H.-S.K. and S.-M.S. ; supervision, S.-M.S. All authors have read and agreed to the published version of the manuscript.

Funding: This research received no external funding.

Institutional Review Board Statement: The study was conducted according to the guidelines of the Declaration of Helsinki, and approved by the Institutional Review Board of Yeungnam university hospital (10-01-031).

Informed Consent Statement: Informed consents were obtained from the patientses arents.

Data Availability Statement: All data are provided either in the paper.

Conflicts of Interest: The authors declare no conflict of interest.

  1. Mention the type of study design.
  2. Mention the study duration and study setting.

Answer: Thank you for your comment. I revised the manuscript as follows.

Ninety-two pediatric patients (mean age, 7.03±2.56 years; range, 2–12 years) were recruited retrospectively according to the following criteria from May 2017 to the end of 2020:

  1. Include the character of the study participants in detail.

Answer: Thank you for your comment. We revised the patients section as follows.

Ninety-two pediatric patients (mean age, 7.03±2.56 years; range, 2–12 years) were recruited retrospectively according to the following criteria from May 2017 to the end of 2020: (1) patients with hemiplegic CP diagnosed by two pediatric neurologists; (2) diagnosis of hemiplegic CP for at least one year; (3) limb length measurement was performed within one month of DTI scanning; (4) absence of congenital anomaly, fracture, and operation history on extremities; and (5) no history of botulinum toxin injection on extremities within the previous 6 months. Patients diagnosed with genetic syndromes, malnutrition, peripheral neuropathy, or endocrinopathies were excluded. The study population was originally selected from 196 pediatric patients with hemiplegic CP, aged 2–12 years. Of these 196 patients, 113 who underwent limb length measurement within one month of DTT scanning were selected. Of these 113 patients, seven were excluded due to a history of botulinum toxin injection within the previous 6 months. Twelve patients with a history of orthopedic surgery of the extremities were excluded. Two patients with genetic syndromes were excluded. The remaining 92 patients were enrolled in the study. (Figure 1). All participants were diagnosed with spastic hemiplegic CP. To reflect the difference in growth-related LLD, the patients who could walk, even with the use of orthoses or walker, were enrolled in the study. Patients with preterm birth < 37 weeks, extremely low birth weight < 1000 g, or brain lesion on conventional MRI were also included. An explanation sheet was provided in consideration of the characteristics of the pediatric patient, and if the patient did not understand the explanation, explanation and informed consent statements were provided to the guardians. All examination procedures and possible side effects and safety issues were explained before the examination. Pediatric neurologists and guardians accompanied the patient throughout the entire process of the study, observing and monitoring the patient’s condition. When the patient’s condition was unfavorable or safety-related issues were suspected, the study was stopped or delayed for the patient. However, none of the 92 patients stopped the test or showed any serious safety-related issues. The pediatric neurologists were unware of the DTT results. Two analysts of DTT (Son SM and Kim HS) were also unaware of the clinical information. They analyzed the DTT in a blind state to the clinical information before the statistical analysis was completed. Informed consent was obtained from the parents of all the participants, and the institutional review board of our hospital approved the study protocol.

  1. Include the ethical and clinical trial registration number.

Answer: I appreciate your comments. It seems that IRB information is missing in the submitted text. I added ethical information in the text.

Institutional Review Board Statement: The study was conducted according to the guidelines of the Declaration of Helsinki, and approved by the Institutional Review Board of Yeungnam university hospital (10-01-031).

  1. Include the randomization and allocation of subjects in detail.

Answer: This study was performed retrospectively. Therefore, no randomization or allocation procedure was performed. First, 196 hemiplegic CP were enrolled and 83 patients who failed to complete DTT and limb length evaluation within one month. Of these 113 patients, 7 patients with history of botulinum toxin injection within the previous 6 months, 12 patients with a history of OS surgery, and 2 patients with genetic syndrome were excluded according to the inclusion/ exclusion criteria. Finally, 92 patients were recruited. After obtaining written consent from the guardian of the patient who wanted to participate in the study, clinical evaluation and DTT were performed. According to the results of CST disruption, group A and group B were divided and compared. A graph has been added to help the understanding of the research method.

A figure has been added to help the understanding of the research process.

  1. How the safety concerns are measured?

Answer: The pediatric neurologists and guardians accompanied the patient throughout the entire process of the study, observing the patient's condition. When the patient’s condition was unfavorable or safety- related issues were suspected, the study for the patient was stopped or delayed. However, none of the 92 patients stopped the test or show any serious safety-related issues.

I added about the safety concerns of ours in the participants section as follows.

…All participants were diagnosed with spastic hemiplegic CP. To reflect the difference in growth-related LLD, the patients who could walk, even with the use of orthoses or walker, were enrolled in the study. Patients with preterm birth < 37 weeks, extremely low birth weight < 1000 g, or brain lesion on conventional MRI were also included. An explanation sheet was provided in consideration of the characteristics of the pediatric patient, and if the patient did not understand the explanation, explanation and informed consent statements were provided to the guardians. All examination procedures and possible side effects and safety issues were explained before the examination. Pediatric neurologists and guardians accompanied the patient throughout the entire process of the study, observing and monitoring the patient’s condition. When the patient’s condition was unfavorable or safety-related issues were suspected, the study was stopped or delayed for the patient. However, none of the 92 patients stopped the test or showed any serious safety-related issues. The pediatric neurologists were unware of the DTT results. Two analysts of DTT (Son SM and Kim HS) were also unaware of the clinical information. They analyzed the DTT in a blind state to the clinical information before the statistical analysis was completed. Informed consent was obtained from the parents of all the participants, and the institutional review board of our hospital approved the study protocol.

  1. Include the study outcome measures and its reliability and validity.

Answer: I appreciate your comments. I added the results of reliability and validity in clinical evaluation and DTT sections as follows.

Two pediatric neurologists examined the patients independently twice with 1-week interval between examinations, and all patients were examined twice by two neurologists. The results of FxL, MAS, and MMT by two neurologists were the same in all patients. The consistent rate of LLD was 96% for the inter-observer’s analysis and 97% for the intra-observer’s analysis when lengths < 1 cm are truncated and limb length was measured in centimeters. These results showed a high reliability and validity rate

To measure the inter- and intra-observer’s variation, the DTT analysis was performed blindly and independently by two authors (Son SM and Kim HS). Each author analyzed the data of each patient twice, randomly, with 1-week interval between examinations. The DTI data of each subject were included as DTT data when the degree of CST disruption result of each analyzer was the same. All results were the same, and all data from the 92 subjects were included as DTT data. The consistent rate of the inter-observer’s analysis was 97%. The consistent rate of the intra-observer’s analysis was 98%, demonstrating high reproducibility rate.

  1. Include the information about the blinding procedures. 

Answer: Thank you for your comment. This study is a retrospective study and did not undergo a special process for blinding or randomization. However, the pediatric neurologist who performed clinical evaluation and two analysts of DTT conducted the examination blindly on each other's test results. We added this in the method section.

The pediatric neurologists were unware of the DTT results. Two analysts of DTT (Son SM and Kim HS) were also unaware of the clinical information. They analyzed the DTT in a blind state to the clinical information before the statistical analysis was completed.

  1. Include the reference study for sample size calculation.

Answer: As mentioned in the previous question, this is a retrospective study which does not require sample size calculation.

  1. The statistical tests used for the study was not apt to this study.

Answer: I totally agreed with your opinion. We have some mistakes in our description. So, we revised as follows.

SPSS software (v.18.0; SPSS, Chicago, IL, USA) was used for data analysis. Independent t-test and the chi-square test were used to determine statistical differences between groups A and B in demographic data. Independent t-test and ANCOVA were used to determine the difference in LLD. The Pearson correlation test was used to determine the statistical significance of the correlation between the LLD and DTI parameters. Statistical significance was set at a P-value<0.05.

  1. The results should be presented with 95%CI (upper limit – lower limit) for all the variables.

Answer: Thank you for your comment. We revised the table 1 indicating difference of 95% CI.

  1. Avoid abbreviations in the conclusion.
  2. The conclusion should be more concise and self-explanatory and drawn on the basis of study reports. 
  3. Add more real-time limitations faced by the researcher and the study. 
  4. Include future recommendations of the study.

(17~20) Answer: We revised the conclusion part as your comment. I appreciate your valuable comment.

In conclusion, the corticospinal tractis thought to have meaningful relevance with distal limb length discrepancy, especially with hand length in patients with hemiplegic cerebral palsy. Therefore, if patients with hemiplegia had severe disruption of CST, close monitoring for the possibility of limb length discrepancy is recommended. To the best of our knowledge, this study is the first to demonstrate the relationship between the LLD and CST in hemiplegic CP using DTT. However, this study has several limitations. First, additional imaging evaluations, such as radiography, were not performed. Second, additional electrophysiological evaluations, including electromyography/nerve conduction velocity or motor-evoked potential tests, were not performed. Third, more detailed identical functional evaluations were performed due to the wide age range of the patients. Fourth, evaluations on the various possibilities of LLD by age and different causes of hemiplegia have not been conducted. Another limitation was lack of healthy control group, so the comparison of LLD between the healthy controls and CP was not evaluated. Finally, the limitations of DTI should be considered. DTI may underestimate or overestimate the neural fiber tract because of the region of fiber complexity. Crossing can prevent the full reflection of the underlying fiber architecture using DTI. Besides, the probabilistic technique using FSL can provide more accurate information about the presence of branching fibers or preservation of weak white matter integrity than the deterministic technique using PRIDE or DTI studio, which was used in this study [24]. Therefore, complementary probabilistic DTI studies with larger case numbers that include participants with different causes of hemiplegia, various ages, and normal healthy subjectsand detailed and identical clinical, radiological, and electrophysiological evaluations are warranted.

Reviewer 2 Report

This study investigated the relationship between limb length discrepancy and corticospinal tract disruption in patients with hemiplegic cerebral palsy. The manuscript is generally written well, yet below I have some concerns and suggestions to improve the manuscript.

There seem to be some misuses between the “limb length (LL)” and “limb length discrepancy (LLD)” throughout the manuscript. For example, there are descriptions of “FA and MD of the affected CSTs were measured and correlated with LLD…”; and “Hand LL was significantly correlated with FA, MD…”. Please check whether the “LL” in the latter sentence should be corrected to “LLD”.

All participants in the current study are children (2–12 years old). Are the data preprocessing pipelines used in this study suitable for children? Please clarify.

In line 203, “additional imaging evaluations, such as X-ray or MRI, were not performed” – I think such statement is inappropriate because DTI is actually a special MRI technique.

“A paired t-test was used to determine statistical differences between groups A and B in demographic data and LLD” – please check if there is a mistake here. How to perform paired t-tests when the number of subjects was not matched between the two groups?

In the statistical analyses -- did the author perform any corrections across multiple tests (e.g., FDR corrections)? Did the authors control possible effects of sex and age?

All the references in this manuscript were published in 2016 or earlier. I suggest the author to refer to some most recent studies in this field. Specially, the authors used deterministic tractography in this study; however, probabilistic tractography has been suggested to be more accurate than deterministic tractography in recent years. This should be considered as a limitation and some recent studies using deterministic tractography should be cited, such as: https://www.frontiersin.org/articles/10.3389/fpsyt.2018.00391/full.

Furthermore, I think that the lack of a healthy control group should be considered as another limitation of this study.

There are some obvious errors in typing and reference format, which should be corrected (e.g., there is an additional space in line 163; and errors in Reference list in lines 235-237). The authors should carefully check the manuscript before publication.

Author Response

This study investigated the relationship between limb length discrepancy and corticospinal tract disruption in patients with hemiplegic cerebral palsy. The manuscript is generally written well, yet below I have some concerns and suggestions to improve the manuscript.

  1. There seem to be some misuses between the “limb length (LL)” and “limb length discrepancy (LLD)” throughout the manuscript. For example, there are descriptions of “FA and MD of the affected CSTs were measured and correlated with LLD…”; and “Hand LL was significantly correlated with FA, MD…”. Please check whether the “LL” in the latter sentence should be corrected to “LLD”.

    Answer: I totally agreed with your opinion. We revised the manuscript as follows.

The degree of LLD between the affected and unaffected side is shown in Table 2. A significant difference was observed only in hand length (p<0.05), not in total upper limb length, total lower limb length, and foot length (p>0.05) from the ANCOVA test. When CST disruption was severe, decreased FA and increased MD were observed, the correlation between these DTI parameters and the degree of LLD showed significant results with FA (r= -0.578) and MD values (r= 0.512) in hand length discrepancy. Total upper, total lower, and foot length discrepancy showed no significant results. The correlation between the degrees of LLD and the degree of CST disruption also revealed no significant results in upper total limb length, lower total limb length, and foot length; however, correlations in the hand length discrepancy (r=-0.721) were statistically significant.

  1. All participants in the current study are children (2–12 years old). Are the data preprocessing pipelines used in this study suitable for children? Please clarify.

Answer: I appreciate your comment. This study was a study designed for pediatric patients, which has been pre-examined by pediatric neurologists with the approval of our hospital’s IRB. An explanation sheet was provided in consideration of the characteristics of the pediatric patient, and if the patient did not understand the explanation, explanation and informed consent statements were provided to the guardians. All examination procedures and possible side effects and safety issues were explained before examination. A pediatric neurologists and guardians accompanied the patient throughout the entire process of the study, observing and monitoring the patient's condition. When the patient’s condition was unfavorable or safety- related issues were suspected, the study for the patient was stopped or delayed. However, none of the 92 patients stopped the test or show any serious safety-related issues. The manuscript was revised in the patient section.

…All participants were diagnosed with spastic hemiplegic CP. To reflect the difference in growth-related LLD, the patients who could walk, even with the use of orthoses or walker, were enrolled in the study. Patients with preterm birth < 37 weeks, extremely low birth weight < 1000 g, or brain lesion on conventional MRI were also included. An explanation sheet was provided in consideration of the characteristics of the pediatric patient, and if the patient did not understand the explanation, explanation and informed consent statements were provided to the guardians. All examination procedures and possible side effects and safety issues were explained before the examination. Pediatric neurologists and guardians accompanied the patient throughout the entire process of the study, observing and monitoring the patient’s condition. When the patient’s condition was unfavorable or safety-related issues were suspected, the study was stopped or delayed for the patient. However, none of the 92 patients stopped the test or showed any serious safety-related issues. The pediatric neurologists were unware of the DTT results. Two analysts of DTT (Son SM and Kim HS) were also unaware of the clinical information. They analyzed the DTT in a blind state to the clinical information before the statistical analysis was completed. Informed consent was obtained from the parents of all the participants, and the institutional review board of our hospital approved the study protocol.

  1. In line 203, “additional imaging evaluations, such as X-ray or MRI, were not performed” – I think such statement is inappropriate because DTI is actually a special MRI technique.

Answer; I appreciate your comment for the manuscript. I totally agreed with your opinion. I revised the sentence,

  1. “A paired t-test was used to determine statistical differences between groups A and B in demographic data and LLD” – please check if there is a mistake here. How to perform paired t-tests when the number of subjects was not matched between the two groups?

 Answer: I totally agreed with your opinion. We have some mistakes in our description. So, we revised as follows.

SPSS software (v.18.0; SPSS, Chicago, IL, USA) was used for data analysis. Independent t-test and the chi-square test were used to determine statistical differences between groups A and B in demographic data. Independent t-test and ANCOVA were used to determine the difference in LLD. The Pearson correlation test was used to determine the statistical significance of the correlation between the LLD and DTI parameters. Statistical significance was set at a P-value<0.05.

  1. In the statistical analyses -- did the author perform any corrections across multiple tests (e.g., FDR corrections)? Did the authors control possible effects of sex and age?

    Answer: This study is a comparative study between two groups, not an individual comparison. Since it is a comparison between two groups (A and B), rather than three or more groups, it is considered that a post-hoc test is not necessary in this study. However, ANCOVA was performed to correct for the effects of age and sex in comparison of demographic data between two groups.

  1. All the references in this manuscript were published in 2016 or earlier. I suggest the author to refer to some most recent studies in this field. Specially, the authors used deterministic tractography in this study; however, probabilistic tractography has been suggested to be more accurate than deterministic tractography in recent years. This should be considered as a limitation and some recent studies using deterministic tractography should be cited, such as: https://www.frontiersin.org/articles/10.3389/fpsyt.2018.00391/full.

 Answer: I totally agreed to your comment. I added it as another limitation and added additional references.

To the best of our knowledge, this study is the first to demonstrate the relationship between the LLD and CST in hemiplegic CP using DTT. However, this study has several limitations. First, additional imaging evaluations, such as radiography, were not performed. Second, additional electrophysiological evaluations, including electromyography/nerve conduction velocity or motor-evoked potential tests, were not performed. Third, more detailed identical functional evaluations were performed due to the wide age range of the patients. Fourth, evaluations on the various possibilities of LLD by age and different causes of hemiplegia have not been conducted. Another limitation was lack of healthy control group, so the comparison of LLD between the healthy controls and CP was not evaluated. Finally, the limitations of DTI should be considered. DTI may underestimate or overestimate the neural fiber tract because of the region of fiber complexity. Crossing can prevent the full reflection of the underlying fiber architecture using DTI. Besides, the probabilistic technique using FSL can provide more accurate information about the presence of branching fibers or preservation of weak white matter integrity than the deterministic technique using PRIDE or DTI studio, which was used in this study [24]. Therefore, complementary probabilistic DTI studies with larger case numbers that include participants with different causes of hemiplegia, various ages, and normal healthy subjectsand detailed and identical clinical, radiological, and electrophysiological evaluations are warranted.

[7]   Schroeder KM, Heydemann JA, Beauvais DH. Musculoskeletal Imaging in Cerebral Palsy. Phys Med Rehabil Clin N Am. 2020 Feb;31(1):39-56.

[8]  Lee JS, Choi IJ, Shin MJ, Yoon JA, Ko SH, Shin YB. Bone age in unilateral spastic cerebral palsy: is there a correlation with hand function and limb length? J Pediatr Endocrinol Metab. 2017 Mar 1;30(3):337-341.

[24]  Long Y, Ouyang X, Liu Z, Chen X, Hu X, Lee E, Chen EYH, Pu W, Shan B, Rohrbaugh RM. Associations Among Suicidal Ideation, White Matter Integrity and Cognitive Deficit in First-Episode Schizophrenia. Front Psychiatry. 2018 Aug 28;9:391.

  1. Furthermore, I think that the lack of a healthy control group should be considered as another limitation of this study

Answer: I agreed with your opinion. I added it as another limitation according to your comment.

To the best of our knowledge, this study is the first to demonstrate the relationship between the LLD and CST in hemiplegic CP using DTT. However, this study has several limitations. First, additional imaging evaluations, such as radiography, were not performed. Second, additional electrophysiological evaluations, including electromyography/nerve conduction velocity or motor-evoked potential tests, were not performed. Third, more detailed identical functional evaluations were performed due to the wide age range of the patients. Fourth, evaluations on the various possibilities of LLD by age and different causes of hemiplegia have not been conducted. Another limitation was lack of healthy control group, so the comparison of LLD between the healthy controls and CP was not evaluated. Finally, the limitations of DTI should be considered. DTI may underestimate or overestimate the neural fiber tract because of the region of fiber complexity. Crossing can prevent the full reflection of the underlying fiber architecture using DTI. Besides, the probabilistic technique using FSL can provide more accurate information about the presence of branching fibers or preservation of weak white matter integrity than the deterministic technique using PRIDE or DTI studio, which was used in this study [24]. Therefore, complementary probabilistic DTI studies with larger case numbers that include participants with different causes of hemiplegia, various ages, and normal healthy subjectsand detailed and identical clinical, radiological, and electrophysiological evaluations are warranted.

  1. There are some obvious errors in typing and reference format, which should be corrected (e.g., there is an additional space in line 163; and errors in Reference list in lines 235-237). The authors should carefully check the manuscript before publication.

Answer: Sorry for that. We revised the manuscript.

Round 2

Reviewer 1 Report

Dear authors,

I really appreciate all your efforts to positively address all the comments raised by me. Now the scientific quality of the article is well improved and eligible for publication in the current status. Regards.

Reviewer 2 Report

N/A